# Maternal and child gluten intake and association with type 1 diabetes: The Norwegian Mother and Child Cohort Study

Nicolai A. Lund-Blix[1,2]*, German Tapia[1], Karl Mårild[1,3], Anne Lise Brantsaeter[4], Pål R. Njølstad[5,6], Geir Joner[2,7], Torild Skrivarhaug[2,7], Ketil Størdal[1,8©], Lars C. Stene[1©]

1 Department of Chronic Diseases and Ageing, Norwegian Institute of Public Health, Oslo, Norway, 2 Department of Paediatric and Adolescent Medicine, Oslo University Hospital, Oslo, Norway, 3 Department of Paediatrics, The Sahlgrenska Academy at University of Gothenburg and Queen Silvia Children's Hospital, Gothenburg, Sweden, 4 Department of Environmental Exposure and Epidemiology, Norwegian Institute of Public Health, Oslo, Norway, 5 Department of Paediatric and Adolescent Medicine, Haukeland University Hospital, Bergen, Norway, 6 KG Jebsen Center for Diabetes Research, Department of Clinical Science, University of Bergen, Bergen, Norway, 7 Institute of Clinical Medicine, University of Oslo, Oslo, Norway, 8 Department of Paediatrics, Østfold Hospital Trust, Grålum, Norway

© These authors contributed equally to this work.
* nicolai.andre.lund-blix@fhi.no

**Data Availability Statement:** Codes for the statistical analyses are available upon request. Access to data can be obtained by sending an application to the Norwegian Mother and Child

## Abstract

### Background

The relationship between maternal gluten intake in pregnancy, offspring intake in childhood, and offspring risk of type 1 diabetes has not been examined jointly in any studies. Our aim was to study the relationship between maternal and child intake of gluten and risk of type 1 diabetes in children.

### Methods and findings

We included 86,306 children in an observational nationwide cohort study, the Norwegian Mother and Child Cohort Study (MoBa), with recruitment from 1999 to 2008 and with follow-up time to April 15, 2018. We used registration of type 1 diabetes in the Norwegian childhood diabetes registry as the outcome. We used Cox proportional hazard regression to estimate hazard ratios (HRs) for the mother's intake of gluten up to week 22 of pregnancy and offspring gluten intake when the child was 18 months old. The average time followed was 12.3 years (0.70–16.0). A total of 346 children (0.4%) children were diagnosed with type 1 diabetes, resulting in an incidence rate of 32.6/100,000 person-years. Mean gluten intake per day was 13.6 g for mothers and 8.8 g for children. There was no association between the mother's intake of gluten in pregnancy and offspring type 1 diabetes, with an adjusted HR (aHR) of 1.02 (95% confidence interval [CI] 0.73–1.43, $p = 0.91$) for each 10-g-per-day increment. There was an association between offspring intake of gluten and a higher risk of type 1 diabetes, with an aHR of 1.46 (95% CI 1.06–2.01, $p = 0.02$) for each 10-g-per-day increment. Among the limitations are the likely imprecision in estimation of gluten intake and that we only had information regarding gluten intake at 2 time points in early life.

Cohort Study (email: dataaccess@fhi.no, information website: https://www.fhi.no/en/op/data-access-from-health-registries-health-studies-and-biobanks/data-from-moba/moba-research-data-files/).

**Funding:** The Norwegian Mother and Child Cohort Study is supported by the Norwegian Ministry of Health and Care Services and the Ministry of Education and Research, NIH/NIEHS (contract no. N01-ES-75558), NIH/NINDS (grant no. 1: UO1 NS 047537-01 and grant no. 2: UO1 NS 047537-06A1). The sub-study was funded by a research grant from the Research Council of Norway (grant 2210909/F20, to LCS). NAL was supported by a grant from Helse Sør-Øst, Norway. KS was supported by an unrestricted grant from Oak Foundation, Geneva, Switzerland. PRN was supported by grants from the Norwegian Research Council (#240413) and Helse Vest (Strategic Grant PERSON-MED-DIA, and #12270). The funders had no role in study design, data collection and analysis, decision to publish, or preparation of the manuscript.

**Competing interests:** The authors have declared that no competing interests exist.

**Abbreviations:** aHR, adjusted hazard ratio; BMI, body mass index; CI, confidence interval; HR, hazard ratio; MoBa, Norwegian Mother and Child Cohort Study; RCT, randomised controlled trial; TEDDY, The Environmental Determinants of Diabetes in the Young.

## Conclusions

Our results show that, while the mother's intake of gluten in pregnancy was not associated with type 1 diabetes, a higher intake of gluten by the child at an early age may give a higher risk of type 1 diabetes.

## Author summary

### Why was this study done?

- Results from another large Nordic cohort showed that the mother's intake of gluten in pregnancy was associated with offspring risk of type 1 diabetes, but the study did not have information on the child's diet.
- To our knowledge, the relationship between maternal gluten intake in pregnancy, offspring intake in childhood, and risk of type 1 diabetes has not been examined jointly in any studies.

### What did the researchers do and find?

- We studied the relationship between maternal and offspring intake of gluten and risk of type 1 diabetes in children from a Norwegian observational nationwide cohort study with recruitment during the period from 1999 to 2008 including more than 80,000 children.
- There was an association between the child's intake of gluten and a higher risk of type 1 diabetes in the child, but not between maternal intake and risk.

### What do these findings mean?

- Our results show that, while the mother's intake of gluten in pregnancy is not associated with type 1 diabetes, a higher gluten intake by the child at an early age may give a higher risk.
- These findings may motivate further research in the field, ideally a randomised controlled trial (RCT).

## Introduction

Type 1 diabetes is an increasingly common disease often presenting in childhood, with some of the highest incidence rates in the Nordic countries [1]. Type 1 diabetes occurs after an immune-mediated destruction of pancreatic beta cells, which in the end leads to lifelong dependence on insulin treatment [2]. While genetic factors are important in the aetiology of the disease, increased incidence rates over recent decades suggest a role for nongenetic factors [1].

Gluten has been hypothesised to be one of the environmental factors involved in the development of type 1 diabetes [3]. It has been found in animal models and in vitro studies that gluten may have an effect on the immune system by increasing proinflammatory cytokine production or may lead to dysbiosis of the gut microbiota [3].

Most of the prospective studies in humans that have examined aspects of gluten intake as a risk factor for type 1 diabetes have focused on age at introduction of cereals or gluten-containing foods in infancy in high-risk cohorts, with inconsistent results [4–11]. Two small studies reported no association between maternal intake of gluten-containing foods in pregnancy and development of islet autoimmunity [12, 13], but a recent, large cohort study found that the mother's intake of gluten while pregnant may increase the probability of the offspring having type 1 diabetes [14]. The relationship between maternal gluten intake in pregnancy, the child's intake at 18 months, and the child's risk of type 1 diabetes have not been examined together in any studies.

The aim was to study the relationship between maternal and child intake of gluten and risk of type 1 diabetes in children.

## Methods

### Participants and study design

We included participants in the Norwegian Mother and Child Cohort Study (MoBa) [15]. MoBa is a prospective nationwide, population-based pregnancy cohort in Norway during 1999–2009. The women consented to participation in 41% of the pregnancies. A total of 86,306 children, of whom 346 developed type 1 diabetes, were included in the analysis (Fig 1).

The current study is based on version VIII of the quality-assured data files. We used a prospective analysis plan in designing the study (S1 Text). We used exposure information from questionnaires (for English versions, see https://www.fhi.no/en/studies/moba/for-forskere-artikler/questionnaires-from-moba) at pregnancy week 22 and offspring age 18 months. Data were linked to the Medical Birth Registry of Norway and the Norwegian Patient Register using personal identity numbers assigned to all Norwegian residents.

### Exposures: Gluten intake during pregnancy and in the child at age 18 months

Intake of gluten was assessed from a semiquantitative food frequency questionnaire completed around pregnancy week 22 and from a questionnaire at child's age 18 months (S1 Fig). For the food frequency questionnaire completed around pregnancy week 22, we derived the average protein intake (grams per day) from gluten-containing flour or grains from the MoBa food database. The food frequency questionnaire covered the period up to week 22 of pregnancy and has been validated for food and nutrient intake [16]. To estimate the child's gluten intake, we used a questionnaire completed at 18 months covering the frequency of wheat-, rye-, and barley-containing food intake. Portion sizes were obtained from a published report [17] and product labels. We estimated the average protein intake (grams per day) from gluten-containing flour or grains by using recipes and nutritional contents of food items from the Norwegian Food Composition Table [18] in addition to traditional recipes and ingredient lists from product labels. The average gluten intake (grams per day) was estimated by using a conversion factor of 0.75 in accordance with most studies [19–21]. The conversion factor of 0.75 is based on conversion factors of 0.80 for wheat, 0.65 for rye, and 0.50 for barley [21–23], wheat being the most widely used grain in the different food products assessed.

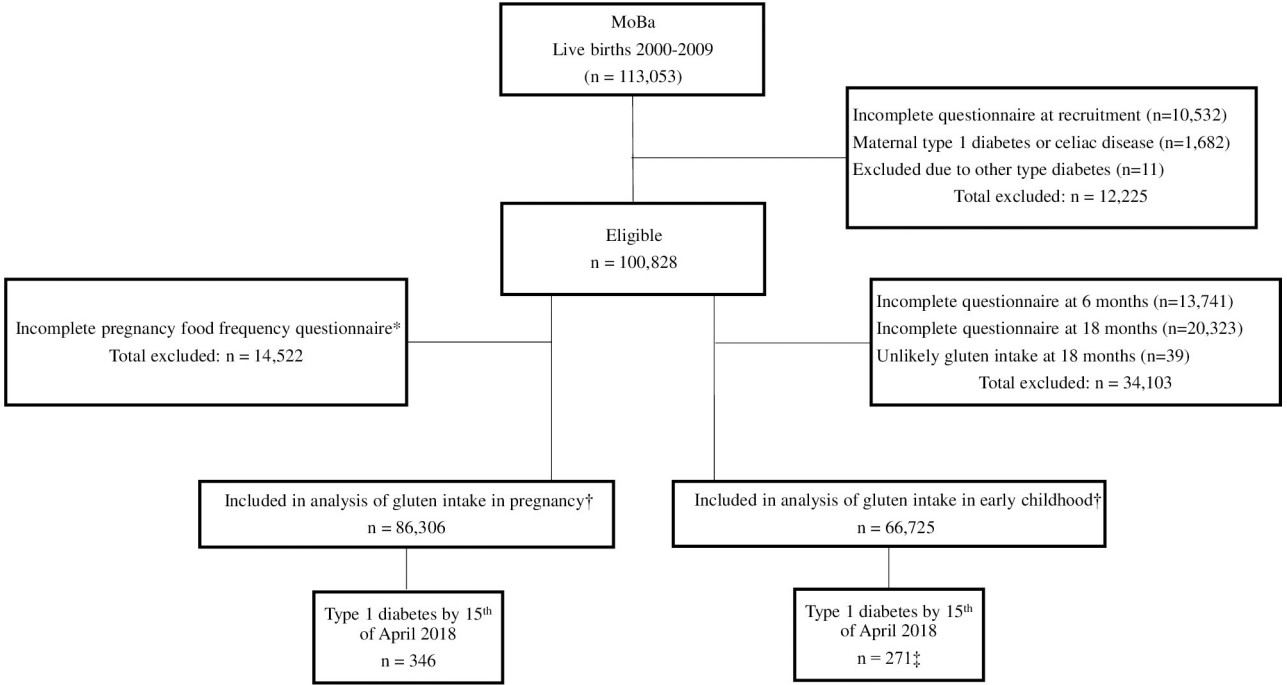

**Fig 1. Flowchart of the study cohort.** *The food frequency questionnaire was introduced in 2002 (response rate 92%). †In the analysis including both pregnancy and childhood gluten intake, there were 62,900 participants with valid data (255 of these were type 1 diabetes cases). ‡Nine cases were excluded from the analyses due to having the outcome of type 1 diabetes prior to the exposure measurement at 18 months. MoBa, Norwegian Mother and Child Cohort Study.

We excluded from the analyses children with a gluten intake above 35 g/day at 18 months of age (*n* = 39, Fig 1). We categorised intake of gluten into percentiles (<10, 10–20, 20–50, 50–80, 80–90, and ≥90) for comparison with the recently published study from Denmark [14]. Characteristics of participants are shown in Table 1. (Characteristics for participants with valid information on intake of gluten in childhood are shown in S1 Table) Characteristics of participants excluded due to missing exposure data are shown in S2 Table.

## Outcome: Type 1 diabetes

Time to clinical type 1 diabetes diagnosis in the child was used as the outcome. Data on offspring type 1 diabetes were obtained from the Norwegian Patient Registry and Norwegian Childhood Diabetes Registry [24]. Nine cases of monogenic or type 2 diabetes were excluded from the study. Identified during follow-up were a total of 346 children with type 1 diabetes with valid information on maternal gluten intake at week 22 of pregnancy and 271 children with valid information on child gluten intake at 18 months of age (Fig 1). Participants excluded due to missing exposure information did not differ in their incidence of type 1 diabetes compared to those included in the current analysis (S2 Fig).

## Other variables

We obtained maternal age, parity, and mode of delivery and child sex, birth weight, and gestational age—categorised as shown in Table 1—from the Medical Birth Registry of Norway. The MoBa recruitment questionnaire completed at week 18 of pregnancy provided data on maternal education, smoking, prepregnant body mass index (BMI), maternal coeliac disease diagnosis, and maternal type 1 diabetes diagnosis. From MoBa questionnaires completed at child age

**Table 1. Characteristics of the study participants included in the analysis of maternal gluten intake during pregnancy and risk of type 1 diabetes in the child.**

| Gluten intake by percentile | All participants | <10% | 10%–20% | 20%–50% | 50%–80% | 80%–90% | >90% |
|---|---|---|---|---|---|---|---|
| Median (range) in g/day | 13.0 (0–62.5) | 5.9 (0–7.6) | 8.7 (7.6–9.5) | 11.4 (9.5–13.0) | 14.8 (13.0–17.3) | 18.4 (17.3–20.1) | 23.0 (20.1–62.5) |
| **Maternal characteristics** | n = 86,306 | n = 8,518 | n = 8,616 | n = 25,962 | n = 25,928 | n = 8,653 | n = 8,629 |
| **Age in years, mean (SD)** | 30.2 (4.6) | 29.7 (4.9) | 30.2 (4.6) | 30.4 (4.5) | 30.5 (4.4) | 30.2 (4.5) | 29.6 (4.9) |
| <25 | 9,165 (10.6) | 1,293 (15.2) | 965 (11.2) | 2,422 (9.3) | 2,267 (8.7) | 924 (10.7) | 1,294 (15.0) |
| 25–34 | 61,812 (71.6) | 5,818 (68.3) | 6,091 (70.7) | 18 797 (72.4) | 18,844 (72.7) | 6,254 (72.3) | 6,008 (69.6) |
| ≥35 | 15,329 (17.8) | 1,407 (16.5) | 1,560 (18.1) | 4,743 (18.3) | 4,817 (18.6) | 1,475 (17.0) | 1,327 (15.4) |
| **Prepregnancy BMI, mean (SD)** | 24.1 (4.3) | 24.3 (4.5) | 24.3 (4.4) | 24.2 (4.3) | 24.0 (4.2) | 23.7 (4.1) | 23.9 (4.4) |
| <20 | 10,522 (12.2) | 1,068 (12.5) | 945 (11.0) | 2,954 (11.4) | 3,133 (12.1) | 1,208 (14.0) | 1,214 (14.1) |
| 20–25 | 47,032 (54.5) | 4,307 (50.6) | 4,672 (54.2) | 14 182 (54.6) | 14,388 (55.5) | 4,838 (55.9) | 4,645 (53.8) |
| 25–29.9 | 18,475 (21.4) | 1,936 (22.7) | 1905 (22.1) | 5,671 (21.8) | 5,496 (21.2) | 1,757 (20.3) | 1,710 (19.8) |
| ≥30 | 8,055 (9.3) | 916 (10.8) | 896 (10.4) | 2,500 (9.6) | 2,289 (8.8) | 646 (7.5) | 808 (9.4) |
| *Missing data* | 2,222 (2.6) | 291 (3.4) | 198 (2.3) | 655 (2.5) | 622 (2.4) | 204 (2.4) | 252 (2.9) |
| **Prematurity** | 5,391 (6.2) | 567 (6.7) | 551 (6.4) | 1,587 (6.1) | 1,540 (5.9) | 550 (6.4) | 596 (6.9) |
| *Missing data* | 61 (0.1) | 6 (0.1) | 7 (0.1) | 22 (0.1) | 16 (0.1) | 3 (0.0) | 7 (0.1) |
| **Parity** | | | | | | | |
| 0 | 39,548 (45.8) | 4,561 (53.5) | 4,196 (48.7) | 11,744 (45.2) | 11,201 (43.2) | 3,858 (44.6) | 3,988 (46.2) |
| 1 | 30,447 (35.3) | 2,667 (31.3) | 2,885 (33.5) | 9,339 (36.0) | 9,561 (36.9) | 3,085 (35.7) | 2,910 (33.7) |
| ≥2 | 16,311 (18.9) | 1,290 (15.1) | 1,535 (17.8) | 4,879 (18.8) | 5,166 (19.9) | 1,710 (19.8) | 1,731 (20.1) |
| **Smoking during pregnancy** | | | | | | | |
| No | 78,332 (90.8) | 7,570 (88.9) | 7,780 (90.3) | 23,814 (91.7) | 23,769 (91.7) | 7,858 (90.8) | 7,541 (87.4) |
| Occasionally | 1,465 (1.7) | 154 (1.8) | 163 (1.9) | 427 (1.6) | 404 (1.6) | 159 (1.8) | 158 (1.8) |
| Yes | 6,044 (7.0) | 745 (8.7) | 619 (7.2) | 1,586 (6.1) | 1,615 (6.2) | 602 (7.0) | 877 (10.2) |
| *Missing data* | 465 (0.5) | 49 (0.6) | 54 (0.6) | 135 (0.5) | 140 (0.5) | 34 (0.4) | 53 (0.6) |
| **Maternal education** | | | | | | | |
| <12 years | 31,116 (36.1) | 3,864 (45.4) | 3,331 (38.7) | 8,858 (34.1) | 8,247 (31.8) | 3,006 (34.7) | 3,810 (44.2) |
| 12–15 years | 34,982 (40.5) | 2,909 (34.2) | 3,408 (39.6) | 10 854 (41.8) | 11,128 (42.9) | 3,537 (40.9) | 3,146 (36.5) |
| ≥16 years | 19,789 (22.9) | 1,672 (19.6) | 1,846 (21.4) | 6,141 (23.7) | 6,442 (24.8) | 2,069 (23.9) | 1,619 (18.8) |
| *Missing data* | 419 (0.5) | 73 (0.9) | 31 (0.4) | 109 (0.4) | 111 (0.4) | 41 (0.5) | 54 (0.6) |
| **Breastfeeding duration, mean (SD)** | 9.9 (4.5) | 9.4 (4.6) | 9.6 (4.6) | 9.9 (4.5) | 10.1 (4.5) | 10.2 (4.5) | 9.6 (4.6) |
| <6.0 months | 12,701 (14.7) | 1,424 (16.7) | 1,395 (16.2) | 3,825 (14.7) | 3,525 (13.6) | 1,139 (13.2) | 1,393 (16.1) |
| 6.0–11.9 months | 26,226 (30.4) | 2,453 (28.8) | 2,596 (30.1) | 8,136 (31.3) | 7,969 (30.7) | 2,630 (30.4) | 2,442 (28.3) |
| ≥12 months | 24,304 (28.2) | 1981 (23.3) | 2,270 (26.3) | 7,433 (28.6) | 7,802 (30.1) | 2,580 (29.8) | 2,238 (25.9) |
| *Missing data* | 23,075 (26.7) | 2,660 (31.2) | 2,355 (27.3) | 6,568 (25.3) | 6,632 (25.6) | 2,304 (26.6) | 2,556 (29.6) |
| **Fibre intake (g), mean (SD)** | 31.2 (11.9) | 19.4 (8.1) | 23.6 (7.7) | 27.6 (7.1) | 33.1 (7.9) | 38.7 (9.0) | 48.4 (16.7) |
| <20th centile | 15,776 (18.3) | 6,003 (70.5) | 4,016 (46.6) | 4,842 (18.7) | 845 (3.3) | 56 (0.6) | 14 (0.2) |
| 20–40th centile | 16,719 (19.4) | 1,318 (15.5) | 2,431 (28.2) | 8,432 (32.5) | 4,098 (15.8) | 360 (4.2) | 80 (0.9) |
| 40–60th centile | 17,418 (20.2) | 624 (7.3) | 1,209 (14.0) | 6,709 (25.8) | 7,358 (28.4) | 1,235 (14.3) | 283 (3.3) |
| 60–80th centile | 17,931 (20.8) | 325 (3.8) | 633 (7.3) | 4,201 (16.2) | 8,328 (32.1) | 3,099 (35.8) | 1,345 (15.6) |
| 80–100th centile | 18,461 (21.4) | 248 (2.9) | 327 (3.8) | 1,778 (6.8) | 5,299 (20.4) | 3,903 (45.1) | 6,906 (80.0) |
| *Missing data* | 1 (0.0) | - | - | - | - | - | 1 (0.0) |
| **Energy intake (MJ), mean (SD)** | 9.8 (3.2) | 6.9 (2.0) | 7.9 (1.8) | 8.9 (1.9) | 10.3 (2.2) | 11.6 (2.5) | 14.1 (5.4) |
| <20th centile | 15,744 (18.2) | 5,699 (66.9) | 3,701 (43.0) | 5,060 (19.5) | 1,205 (4.6) | 70 (0.8) | 9 (0.1) |
| 20–40th centile | 16,886 (19.6) | 1,527 (17.9) | 2,632 (30.5) | 7,694 (29.6) | 4,408 (17.0) | 517 (6.0) | 108 (1.3) |
| 40–60th centile | 17,420 (20.2) | 703 (8.3) | 1,309 (15.2) | 6,636 (25.6) | 6,833 (26.4) | 1,468 (17.0) | 471 (5.5) |
| 60–80th centile | 17,883 (20.7) | 369 (4.3) | 699 (8.1) | 4,541 (17.5) | 7,804 (30.1) | 2,828 (32.7) | 1,642 (19.0) |
| 80–100th centile | 18,372 (21.3) | 220 (2.6) | 275 (3.2) | 2,031 (7.8) | 5,678 (21.9) | 3,770 (43.6) | 6,398 (74.1) |
| *Missing data* | 1 (0.0) | - | - | - | - | - | 1 (0.0) |
| **Offspring characteristics** | | | | | | | |

*(Continued)*

**Table 1.** (Continued)

| Gluten intake by percentile | All participants | <10% | 10%–20% | 20%–50% | 50%–80% | 80%–90% | >90% |
|---|---|---|---|---|---|---|---|
| Median (range) in g/day | 13.0 (0–62.5) | 5.9 (0–7.6) | 8.7 (7.6–9.5) | 11.4 (9.5–13.0) | 14.8 (13.0–17.3) | 18.4 (17.3–20.1) | 23.0 (20.1–62.5) |
| Maternal characteristics | n = 86,306 | n = 8,518 | n = 8,616 | n = 25,962 | n = 25,928 | n = 8,653 | n = 8,629 |
| Cesarean section | 13,001 (15.1) | 1,450 (17.0) | 1,401 (16.3) | 3,863 (14.9) | 3,730 (14.4) | 1,225 (14.2) | 1,332 (15.4) |
| Type 1 diabetes | 346 (0.4) | 23 (0.3) | 45 (0.5) | 102 (0.4) | 99 (0.4) | 42 (0.5) | 35 (0.4) |
| Female | 42,179 (48.9) | 4,189 (49.2) | 4,201 (48.8) | 12,749 (49.1) | 12,715 (49.0) | 4,211 (48.7) | 4,114 (47.7) |
| Birth weight (g), mean (SD) | 3,561 (587) | 3,514 (593) | 3,539 (587) | 3,561 (585) | 3,580 (583) | 3,571 (585) | 3,557 (598) |
| <2,500 | 3,585 (4.2) | 403 (4.7) | 378 (4.4) | 1,108 (4.3) | 965 (3.7) | 341 (3.9) | 390 (4.5) |
| 2,500–3,499 | 33,185 (38.5) | 3,538 (41.5) | 3,401 (39.5) | 9,934 (38.3) | 9,731 (37.5) | 3,267 (37.8) | 3,314 (38.4) |
| 3,500–4,499 | 45,932 (53.2) | 4,261 (50.0) | 4,529 (52.6) | 13,860 (53.4) | 14,067 (54.3) | 4,690 (54.2) | 4,525 (52.4) |
| ≥4,500 | 3,598 (4.2) | 315 (3.7) | 308 (3.6) | 1,058 (4.1) | 1,162 (4.5) | 355 (4.1) | 400 (4.6) |
| *Missing data* | 6 (0.0) | 1 (0.0) | - | 2 (0.0) | 3 (0.0) | - | - |
| Age at gluten introduction | | | | | | | |
| <4.0 months | 515 (0.6) | 70 (0.8) | 52 (0.6) | 135 (0.5) | 129 (0.5) | 46 (0.5) | 83 (1.0) |
| 4.0–5.9 months | 16,603 (19.2) | 1,575 (18.5) | 1,657 (19.2) | 4,950 (19.1) | 4,974 (19.2) | 1,710 (19.8) | 1,737 (20.1) |
| ≥6 months | 59,347 (68.8) | 5,614 (65.9) | 5,883 (68.3) | 18,193 (70.1) | 18,061 (69.7) | 5,935 (68.6) | 5,661 (65.6) |
| *Missing data* | 9,841 (11.4) | 1,259 (14.8) | 1,024 (11.9) | 2,684 (10.3) | 2,764 (10.7) | 962 (11.1) | 1,148 (13.3) |
| Coeliac disease diagnosis | 828 (1.0) | 70 (0.8) | 84 (1.0) | 253 (1.0) | 269 (1.0) | 83 (1.0) | 69 (0.8) |
| *Missing data* | 153 (0.2) | 16 (0.2) | 22 (0.3) | 55 (0.2) | 33 (0.1) | 12 (0.1) | 15 (0.2) |
| Weight gain (kg) 0–12 months, mean (SD) | 6.4 (1.05) | 6.4 (1.07) | 6.4 (1.07) | 6.4 (1.04) | 6.4 (1.05) | 6.3 (1.06) | 6.4 (1.06) |
| *Missing data* | 28,829 (33.4) | 3,231 (37.9) | 2,923 (33.9) | 8,238 (31.7) | 8,298 (32.0) | 2,911 (33.6) | 3,228 (37.4) |

**Abbreviation:** BMI, body mass index

6 and 18 months, we obtained information on breastfeeding duration and child age at the time of gluten introduction [25]. MoBa age-7 or -8 questionnaires and register linkage to the Norwegian Patient Register (to the end of 2016) provided information on the child's coeliac disease diagnosis [26]. The Norwegian Patient Register also provided data on parents' medical codes indicative of coeliac disease and type 1 diabetes. We excluded children of mothers with type 1 diabetes or coeliac disease due to the influence on the mother's intake of gluten, and potentially the child's intake (n = 1,682; Fig 1).

## Statistical analysis

Based on the previous Danish study [14] showing a relative risk of 2 comparing the upper and lower categories of maternal gluten intake, we calculated that our sample size would provide us with a statistical power of 94% for the analyses of maternal gluten intake and 88% for child's gluten intake.

We used Cox regression analysis to estimate hazard ratios (HRs) with 95% confidence intervals (CIs). Follow-up time was counted from birth (when analysing mother's gluten intake) or from child's age of 18 months (when analysing child's gluten intake) to type 1 diabetes diagnosis or end of follow-up (April 15, 2018). In analyses mutually adjusting for maternal and child gluten intake, we started the follow-up at 18 months. We found no evidence for violation of the proportional hazards assumption by visually assessing log-minus log plots or formally testing Schoenfeld residuals. We used robust cluster variance estimation to account for potential correlation among siblings in the cohort. We predefined statistical significance as *p*-values of 0.05 or 95% CIs for the HR not including 1.00. The primary analysis was further predefined to be the test for linear trend (per 10-g increase in gluten intake per day), with

covariates defined in model 2 (see subsequently), and to be based on a data set in which we imputed missing covariates using multiple imputation with chained equations [27]. To test nonlinearity, we analysed gluten as a categorical variable using the same cut-offs as the Danish national prospective cohort study (Table 1 and S1 Table) [14].

In addition to unadjusted analyses, we decided a priori to adjust for variables that may be associated with gluten intake and type 1 diabetes. Model 1 adjusted for covariates similar to the Danish national prospective cohort study [14]: prepregnancy maternal BMI, age, parity, smoking status, education, breastfeeding duration, cesarean section, energy intake, and child's sex. Model 2, our primary model, additionally adjusted for age at gluten introduction, birth weight, prematurity, weight gain 0–12 months, and maternal fibre intake during pregnancy and mutually adjusted maternal and child's gluten intake. (In the mutually adjusted analyses, maternal gluten intake is a potential confounder for the child's gluten intake–type 1 diabetes association, whereas child's gluten intake is a potential mediator in the maternal gluten intake–type 1 diabetes association.)

In sensitivity analyses to assess the impact of imputing missing covariates, we repeated the main analyses in those with complete covariate data. In response to a peer review comment, we made a Kaplan Meier survival plot of the main analyses stratified by child's sex and birth weight, and we made a density plot showing the distribution of maternal gluten intake in pregnancy and child's gluten intake at age 18 months. We also assessed the association of maternal fibre intake and gluten intake from refined grains during pregnancy and risk of type 1 diabetes in the child. Also, we assessed the impact of further adjusting our analyses for the child's coeliac disease, and we performed analyses after excluding children who were diagnosed with coeliac disease during follow-up, as suggested in a peer review comment. In response to other peer review comments, we did additional analyses including children who have a mother with type 1 diabetes or celiac disease in the primary model, and we additionally adjusted for other types of diabetes in the mother (type 2 diabetes or gestational diabetes obtained from the medical birth registry of Norway) in the primary model. All analyses were done in Stata version 15 (College Station, TX).

## Patient and public involvement

We conducted the study on previously collected data with no patient involvement in development of the research question or outcome measures, nor in the development of design, the recruitment, or the conduct of the study. Patients were not asked to advise on interpretation or writing up of results. Results from MoBa are disseminated to study participants through press releases, a yearly newsletter sent to all participating families, and the study's and the Norwegian Institute of Public Health's websites.

## Ethical approval

The establishment and data collection in MoBa was previously based on a licence from the Norwegian Data Inspectorate and approval from the Regional Committee for Medical and Health Research Ethics South East, and it is now based on regulations related to the Norwegian Health Registry Act. All participants provided written informed consent. The present study was approved by the Regional Committee for Medical and Health Research Ethics South East.

## Results

Of all participating children, 0.4% were diagnosed with type 1 diabetes ($n$ = 346) after a mean of 12.3 years (range 0.7–16.0) of follow-up (Fig 1). The mean age at diagnosis was 7.5 years (range 0.7–15.0). We had information on islet autoantibodies (towards insulin, glutamic acid decarboxylase, and IA2) at diagnosis from 76% the children who developed type 1 diabetes,

and 92% were positive for at least one islet autoantibody. The overall incidence rate of type 1 diabetes per 100,000 person-years was 32.6. Mean maternal gluten intake during pregnancy was 13.6 g/day (standard deviation 5.2, median 13.0 g/day), and the child's mean gluten intake at age 18 months was 8.8 g/day (standard deviation 3.6, median 8.2 g/day). The distribution of the mother's intake of gluten and offspring intake of gluten in childhood is shown in S3 Fig.

Gluten intake was lower ($p < 0.001$) in mothers who had a lower fibre and energy intake (Table 1). Gluten intake was also lower ($p < 0.001$) in children of mothers who were younger and less educated and who breastfed their child for a shorter duration (S1 Table). Children of mothers with a low gluten intake during pregnancy tended to have a lower gluten intake at 18 months, but the correlation was weak (Spearman's rho = 0.11, $p < 0.001$, S4 Fig).

## Maternal gluten intake during pregnancy and risk of type 1 diabetes in the child

Maternal gluten intake during pregnancy was not associated with risk of type 1 diabetes in the child (Table 2). For each 10-g/day-increase of gluten intake, the adjusted hazard ratio (aHR) was 1.02 (95% CI 0.73–1.43, $p = 0.91$). For each standard deviation (5.2 g/day) increase of gluten intake, the aHR was 1.01 (95% CI 0.85–1.20, $p = 0.91$). While there was a tendency towards a nonlinear association—with higher risk of type 1 diabetes in those with the next lowest and in those with the next highest category of maternal gluten intake—the global likelihood ratio test did not support a significant nonlinear association ($p = 0.11$).

## Gluten intake by the child at age 18 months and subsequent risk of type 1 diabetes

The child's intake of gluten was significantly associated with higher risk of type 1 diabetes with a dose-response relationship (Table 2). For each 10-g/day increase of gluten intake, the aHR was 1.46 (95% CI 1.06–2.01, $p = 0.02$). For each standard deviation (3.6 g/day) increase of gluten intake, the aHR was 1.15 (95% CI 1.02–1.29, $p = 0.02$).

While the child's gluten intake remained significantly associated with risk of type 1 diabetes after adjustment for maternal gluten intake during pregnancy, the suggestive association with maternal gluten intake in category 2 and 5 were blunted after adjustment for child's gluten intake (Table 2).

## Additional analyses

Analyses of those with complete covariate data showed similar results as in the main analyses (S3 Table). The overall pattern across subgroups for birth weight and child's sex was consistent with the main findings (S5 Fig). We found no association between maternal dietary fibre intake or gluten intake from refined grains during pregnancy and risk of type 1 diabetes in the child (S4 Table and S5 Table). In analyses with further adjustment for child coeliac disease, our results were essentially unchanged (Table 2). However, since the HR for child's gluten intake was slightly attenuated, we performed another sensitivity analysis after excluding children who were diagnosed with coeliac disease during follow-up. The HR remained similar (aHR = 1.37, 95% CI 0.98–1.93, $p = 0.07$ per 10-g increase for childhood intake), adjusted as in model 2, but it was no longer significant due to the reduction of type 1 diabetes cases. Including children who have a mother with type 1 diabetes or celiac disease in the primary model gave essentially unchanged results (S6 Table). Finally, we additionally adjusted for other types of diabetes in the mother (type 2 diabetes or gestational diabetes obtained from the medical birth registry of Norway) in the primary model, and the results were essentially unchanged

**Table 2. Association between maternal gluten intake during pregnancy (n = 86,306) or the child's intake at 18 months (n = 66,725) and the risk of type 1 diabetes in the child.**

| Gluten intake | Cases | Incidence rate (per 100,000) | HR (95% CI) of type 1 diabetes | | | | | | | |
|---|---|---|---|---|---|---|---|---|---|---|
| | | | Unadjusted | p-Value | Adjusted model 1* | p-Value | Adjusted model 2† (primary model) | p-Value | Adjusted model 3‡ | p-Value |
| **Maternal/pregnancy** | | | | | | | | | | |
| Continuous, per 10-g/day increase | 346 | 36.6 | 1.04 (0.86–1.27) | 0.68 | 1.02 (0.81–1.30) | 0.84 | 1.02 (0.73–1.43) | 0.91 | 1.01 (0.71–1.42) | 0.97 |
| By category | | | | | | | | | | |
| <7.6 g/day | 23 | 22.3 | Ref. | | Ref. | | Ref. | | Ref. | |
| 7.6–9.5 g/day | 45 | 42.6 | 1.90 (1.13–3.18) | 0.01 | 1.90 (1.13–3.20) | 0.01 | 1.71 (0.94–3.09) | 0.08 | 1.65 (0.91–3.00) | 0.10 |
| 9.5–13.0 g/day | 102 | 32.0 | 1.42 (0.89–2.27) | 0.14 | 1.44 (0.90–2.31) | 0.13 | 1.37 (0.79–2.37) | 0.26 | 1.32 (0.76–2.30) | 0.32 |
| 13.0–17.3 g/day | 99 | 31.1 | 1.39 (0.87–2.22) | 0.17 | 1.39 (0.84–2.29) | 0.18 | 1.23 (0.67–2.26) | 0.51 | 1.20 (0.65–2.20) | 0.56 |
| 17.3–20.1 g/day | 42 | 39.5 | 1.76 (1.04–2.96) | 0.03 | 1.81 (1.04–3.16) | 0.04 | 1.50 (0.75–2.98) | 0.25 | 1.41 (0.71–2.82) | 0.33 |
| >20.1 g/day | 55 | 32.7 | 1.45 (0.85–2.49) | 0.18 | 1.44 (0.78–2.63) | 0.23 | 1.60 (0.75–3.39) | 0.22 | 1.56 (0.73–3.33) | 0.24 |
| $p_{trend}$ | | | 0.43 | | 0.45 | | 0.53 | | | 0.55 |
| **Child/18 months of life** | | | | | | | | | | |
| Continuous, per 10-g/day increase | 262 | 35.9 | 1.44 (1.05–1.98) | 0.03 | 1.44 (1.05–1.98) | 0.02 | 1.46 (1.06–2.01) | 0.02 | 1.41 (1.01–1.96) | 0.04 |
| By category | | | | | | | | | | |
| <4.8 g/day | 19 | 26.3 | Ref. | | Ref. | | Ref. | | Ref. | |
| 4.8–5.8 g/day | 25 | 34.3 | 1.32 (0.72–2.44) | 0.37 | 1.32 (0.72–2.43) | 0.37 | 1.30 (0.71–2.40) | 0.39 | 1.37 (0.73–2.56) | 0.32 |
| 5.8–8.2 g/day | 77 | 34.9 | 1.35 (0.81–2.26) | 0.25 | 1.34 (0.80–2.25) | 0.26 | 1.33 (0.80–2.23) | 0.27 | 1.40 (0.82–2.37) | 0.22 |
| 8.2–11.4 g/day | 76 | 34.7 | 1.25 (0.74–2.10) | 0.40 | 1.24 (0.74–2.10) | 0.41 | 1.24 (0.74–2.09) | 0.42 | 1.24 (0.73–2.11) | 0.43 |
| 11.4–13.5 g/day | 30 | 41.2 | 1.61 (0.89–2.90) | 0.11 | 1.60 (0.89–2.89) | 0.12 | 1.61 (0.89–2.91) | 0.11 | 1.58 (0.87–2.89) | 0.14 |
| >13.5 g/day | 35 | 48.2 | 1.84 (1.04–3.27) | 0.04 | 1.85 (1.04–3.28) | 0.04 | 1.86 (1.05–3.30) | 0.03 | 1.86 (1.04–3.34) | 0.04 |
| $p_{trend}$ | | | 0.05 | | 0.05 | | 0.04 | | | 0.07 |

*Model 1 (as in Antvorskov and colleagues [14]): adjusted for maternal age, prepregnant maternal BMI, parity, smoking during pregnancy, education, cesarean section, breastfeeding, sex, and energy intake.

†Model 2 (primary model): as model 1 with additional adjustment for birth weight, age at gluten introduction, prematurity, fibre intake, weight gain 0–12 months, and child's or mother's gluten intake (mutually adjusted exposures).

‡Model 3: as model 2 with additional adjustment for child coeliac disease.

**Abbreviations:** BMI, body mass index; CI, confidence interval; HR, hazard ratio; Ref., Reference category

(HR 1.48, 95% CI 1.09–1.99, p = 0.01, per 10-g/day increase in the child's gluten intake, and HR 0.98, 95% CI 0.71–1.36, p = 0.91, per 10-g/day increase in the maternal gluten intake).

## Discussion

In this study, with both maternal and child gluten intake as exposures, we found that the child's gluten intake at 18 months of age was associated with the risk of type 1 diabetes while maternal gluten intake during pregnancy was not.

## Comparison with previous studies

Our results on the mother's intake of gluten during pregnancy are not in line with those from the only previous study to assess this [14], despite remarkable similarities in characteristics, maternal gluten intake, and methodology. A possible explanation for the discrepancy could be that we were able to exclude participants based on maternal coeliac disease, while the other study neither excluded nor adjusted for this in their analyses. Two studies of high-risk children that investigated maternal intake of gluten-containing foods in pregnancy and development of islet autoimmunity reported no significant association [12, 13]. Of note, both the exposure (intake of gluten-containing cereals rather than amount of gluten) and outcome was different from our study, and the number of children with outcome was small in these high-risk cohorts. A recent study of high-risk children found that an increased gluten intake in childhood was associated with islet autoimmunity but not with type 1 diabetes [28], while another study of high-risk children found no significant association between the child's gluten intake and islet autoimmunity or progression from islet autoimmunity to type 1 diabetes [29]. Again, the apparent inconsistency may have been due to studies investigating different outcomes (clinical type 1 diabetes versus progression) in different populations or due to lack of power. Our results are probably generalizable to other industrialised countries but may not be applicable to populations of non-European origin. In a recently published study from The Environmental Determinants of Diabetes in the Young (TEDDY), a higher gluten intake during the first 5 years of life was associated with increased risk of coeliac disease autoimmunity and coeliac disease among genetically predisposed children [30]. In another cohort of at-risk children, gluten intake between the ages of 1 and 2 years—and, to a lesser extent, the cumulative intake throughout childhood—were associated with the appearance of coeliac disease autoimmunity and coeliac disease [31]. We have previously reported that an increased gluten intake at 18 months was associated with a modestly increased risk of coeliac disease in children from MoBa [32].

## Strengths and weaknesses

The prospective design with recruitment in pregnancy, large sample size, and linkage to national registries with high levels of ascertainment are the strongest advantages of the study. Missing data on child's gluten intake at 18 months of age was mainly due to loss of follow-up, which occurs in all cohorts that are based on voluntary participation. Having no significant difference in risk of type 1 diabetes in those without complete data suggests that missing-data bias was not a serious weakness. Furthermore, multiple imputation analyses have further contributed to minimising any influence of bias due to missing data [27].

The recently published Danish national prospective cohort study has been criticised for not including other dietary components related to gluten intake [33]. We included data on maternal dietary fibre intake and gluten intake from refined grains during pregnancy. We also adjusted for age at gluten introduction and infant weight gain [34]. We are not able to explore all aspects of food intake; however, no dietary factors are established as a risk factor for type 1 diabetes [35]. We accounted for coeliac disease, both by excluding children of mothers with coeliac disease from the analyses and adjusting for child coeliac disease.

Observational studies cannot exclude the possibility of unmeasured, unknown confounders that could have influenced the results. Yet no environmental aetiological factors are clear, plausible confounders.

We assessed gluten intake prospectively, but imprecisions in the estimations are likely, and we only had information regarding gluten intake for the mother in the first half of pregnancy and for the child at 18 months of age and no other time points in early life. We used

questionnaires, which are suitable for estimating regular diet consumed over time but are not as precise as other dietary assessment methods such as a dietary recall or records covering shorter time periods—which, again, do not detect foods consumed outside the time of data registration. The estimated intake of gluten was comparable to what has been reported in studies using the same or other dietary assessment methods, both for adults [14] and children [36–39]. It is unlikely that imprecisions in gluten intake assessment from questionnaires is influenced by future type 1 diabetes. Any bias is nondifferential when we have accounted for coeliac disease in the child and in the mother. The large majority of type 1 diabetes cases were diagnosed long after 18 months, with the potential for reverse causation likely small. We did not have prediagnostic data on islet autoantibodies and are therefore not able to provide information on whether the associations with gluten intake are related to risk of islet autoimmunity or progression from islet autoimmunity to type 1 diabetes. Another limitation is the lack of genetic susceptibility variants. However, while these genetic variants obviously predict type 1 diabetes in the child, it is not obvious that they will confound the associations between maternal or child's gluten intake and type 1 diabetes by influencing also gluten intake, independently of coeliac disease. We did not have detailed information on different ethnicities, but MoBa primarily consists of ethnic Norwegians.

Self-selection in recruitment and losses to follow-up could lead to a nonrepresentative sample in the analyses. Those consenting to participate, and those continuing to participate in MoBa, differ from the average population of women giving birth in Norway, in that the participants have high maternal age and education and there is a low proportion of daily smokers among them [40]. Selection has been shown to play a minor role in association studies in MoBa [40]. Adjusting for maternal education, age, and smoking during pregnancy did not change our results substantially, suggesting that selection bias is unlikely to have had a major influence on our findings.

## Implications for research, clinical practice, and public policy

Our results show that, while the mother's intake of gluten in pregnancy was not associated with type 1 diabetes, a higher intake of gluten by the child at an early age may give a higher risk. The results from a similar Danish national prospective cohort study [14] was not replicated in our cohort, suggesting that there may not be causality. This may also be the case for our finding of childhood intake of gluten as a possible risk factor for type 1 diabetes later in life, which needs to be replicated in other large cohorts or intervention studies.

The child's gluten intake at 18 months of age may possibly increase the risk of type 1 diabetes through several mechanisms related to the immune system [3]. It has been shown that increased gut permeability, which facilitates abnormal absorption of macromolecules, is associated with type 1 diabetes and is detectable before clinical onset [41]. In a recently published study from TEDDY, small changes in abundance of bacterial genera were found in cases with type 1 diabetes compared to controls, such that controls had a higher abundance of bacterial genera that might be indicative of enhanced gut integrity [42]. The child's diet in early life is more important for microbiota development in the child than maternal diet during pregnancy, and our assessment of childhood intake of gluten was during the transitional phase of the developing microbiome, before the stable phase that is observed after 30 months of age [42]. The median age at seroconversion to islet autoimmunity preceding type 1 diabetes is around 24 months of age [43].

If replicated in prospective studies, these findings may motivate further interventional studies of gluten amount in childhood and type 1 diabetes. The results from our study should not lead to changing of recommendations or lead to unnecessary dietary changes among

individuals to avoid type 1 diabetes. Ours and other published data to date should be interpreted cautiously, as there are currently few prospective studies in this field, with conflicting results, and no randomised intervention studies.

## Conclusion

In conclusion, a higher intake of gluten in the child's diet at 18 months, but not in maternal diet during pregnancy, was associated with an increased risk of type 1 diabetes in the child.

## Supporting information

**S1 STROBE Checklist.**
(DOC)

**S1 Table. Characteristics of the study participants included in the analysis of the child's gluten intake at age 18 months and risk of type 1 diabetes.**
(DOCX)

**S2 Table. Characteristics of excluded participants due to missing exposure data.**
(DOCX)

**S3 Table. Association between maternal gluten intake during pregnancy ($n$ = 54,332) or child's intake at 18 months ($n$ = 56,257) and the risk of type 1 diabetes in the child.** Complete case analyses*.
(DOCX)

**S4 Table. Association between maternal intake of fibre during pregnancy and the risk of type 1 diabetes in the child.**
(DOCX)

**S5 Table. Association between maternal intake of gluten from refined grains during pregnancy and the risk of type 1 diabetes in the child***.
(DOCX)

**S6 Table. Analyses of model 2 (primary model) including children who have a mother with type 1 diabetes or coeliac disease ($n$ = 1,033)***.
(DOCX)

**S1 Fig. Excerpts of questionnaires at week 22 of pregnancy and child's age 18 months.**
(DOCX)

**S2 Fig. Cumulative incidence of type 1 diabetes in participants included in the analyses and in those excluded due to missing gluten data.**
(DOCX)

**S3 Fig. Distribution of maternal gluten intake during pregnancy and child's gluten intake at age 18 months.**
(DOCX)

**S4 Fig. Correlation between maternal gluten intake during pregnancy and child's gluten intake at age 18 months.**
(DOCX)

**S5 Fig. Association between maternal gluten intake during pregnancy and child's gluten intake at age 18 months and risk of type 1 diabetes, stratified by child's sex and birth**

**weight.**
(DOCX)

**S1 Text. Analysis plan.**
(DOCX)

## Acknowledgments

We are grateful to all the participating families in Norway who take part in this ongoing cohort study.

## Author Contributions

**Conceptualization:** Ketil Størdal, Lars C. Stene.

**Data curation:** Anne Lise Brantsaeter, Pål R. Njølstad, Geir Joner, Torild Skrivarhaug.

**Formal analysis:** Nicolai A. Lund-Blix, German Tapia, Ketil Størdal, Lars C. Stene.

**Funding acquisition:** Ketil Størdal, Lars C. Stene.

**Investigation:** Nicolai A. Lund-Blix, Ketil Størdal, Lars C. Stene.

**Methodology:** Nicolai A. Lund-Blix, Karl Mårild, Anne Lise Brantsaeter.

**Project administration:** Ketil Størdal, Lars C. Stene.

**Resources:** Anne Lise Brantsaeter, Pål R. Njølstad, Geir Joner, Torild Skrivarhaug, Ketil Størdal, Lars C. Stene.

**Writing – original draft:** Nicolai A. Lund-Blix, German Tapia, Ketil Størdal, Lars C. Stene.

**Writing – review & editing:** Nicolai A. Lund-Blix, German Tapia, Karl Mårild, Anne Lise Brantsaeter, Pål R. Njølstad, Geir Joner, Torild Skrivarhaug, Ketil Størdal, Lars C. Stene.

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
