## [Decision Letter · Decision Letter 0]

31 Oct 2019

Dear Dr. Lund-Blix,

Thank you very much for submitting your manuscript "Maternal and child gluten intake and risk of type 1 diabetes: The Norwegian Mother and Child Cohort Study" (PMEDICINE-D-19-03046) for consideration at PLOS Medicine. 

[LINK]

In light of these reviews, I am afraid that we will not be able to accept the manuscript for publication in the journal in its current form, but we would like to consider a revised version that addresses the reviewers' and editors' comments. Obviously we cannot make any decision about publication until we have seen the revised manuscript and your response, and we plan to seek re-review by one or more of the reviewers. 

We expect to receive your revised manuscript by Nov 14 2019 11:59PM. Please email us (plosmedicine@plos.org) if you have any questions or concerns.

We look forward to receiving your revised manuscript. 

Sincerely,

Clare Stone Acting Editor-in-Chief

for

Louise Gaynor-Brook, MBBS PhD

Associate Editor 

PLOS Medicine

plosmedicine.org

Abstract – please use both 95%Cis and p values for all quantifiable data (also throughout main text and tables, as needed)

Please add a sentence on the study’s limitations as the final sentence of the ‘Methods and Findings’ section of the abstract. 

As this is an observational study and not a trial, please avoid overheated language, such as “could increase” (conclusions in abstract)…”may” would be better (again, line 296)

(please remove the 2 sections what we know / what this study adds)

Please use square brackets in the main text for references and site the refs before the full stop, instead of after. 

Line 235 – “to the best of our knowledge”

Line 344 – please remove the section on copyright (this will be an open access publication). Please also remove funding and competing interests statements etc as these are automatically pulled in from the submission system. 

Strobe – please use sections and paragraph numbers instead of pages as these can alter during revisions / formatting, etc. 

I just wanted to check that all questionnaire material is provided – I see one URL, but there are a couple of mentions of questionnaires. Any additional ones can be provided as Supp files. Apols if they are all provided already. 

Comments from the reviewers:

Reviewer #1: Thank you for the opportunity to review this paper. The authors report findings on a large Norwegian birth cohort on the association between maternal pregnancy gluten intake up to 22 weeks and childhood gluten intake at 18 months and Type 1 diabetes in two separate mutually adjusted Cox regression models. They found that increasing gluten intake in children at 18 months was associated future increased risk of Type 1 diabetes in children but no association between maternal pregnancy gluten intake and risk of Type 1 diabetes in children. This study is well-presented and provides some convincing observational evidence that childhood gluten may have some role to play in increasing risk of type 1 diabetes. 

The adjusted Cox regression models are appropriate for the analyses. The authors have adjusted for confounders/covariates, considered the impact of missing values (completing a complete-case analysis for comparison to the imputed), and excluded patients who may have key confounding factors which may alter the exposure patterns (i.e. mothers with Type 1 diabetes or monogenic diabetes). They have used a cluster variance method to also account for correlation between siblings. I only have a few comments that could be considered.

1) Just a matter of presentation: I would suggest the exposures paragraph (starting lines 116) are presented before the outcome section in the methods as authors mention the exposures comparisons in the outcome section prior to the exposure being described (for instance, lines 111-112 about exclusion due to missing exposure information). 

2) Other variables (Lines 142): Prior history of type 2 diabetes of mothers factored considered and gestational diabetes - as both of these would potentially affect dietary factors, including gluten intake. If possible and information is available, please adjust for both these factors in the regression models (though I would think you could obtain the medical codes from the Norwegian Patient Register). If not, then the discussion limitations need to reflect this. 

3) Statistical analysis (lines 157): There are two primary models presented (1) looking at maternal gluten intake (2) looking at childhood gluten intake. The authors have mutually adjusted each of the gluten intake in both of these models as covariates. Whilst the childhood model makes sense to conduct a baseline adjustment for the maternal gluten intake (as the maternal gluten intake occurs PRIOR to the childhood intake), the maternal model using a baseline adjustment for the childhood gluten intake (which occurs AFTER the exposure period), may not entirely be appropriate. It would be better to incorporate the effects of the childhood intake as a time-dependent covariate in the model instead if you want to stick with a time 0 = birth model. 

The issue with the baseline adjustment of covariates is the effect on survival time. In your maternal model, starting point of follow-up occurs at time 0 = birth. That means that anything occurring afterwards such as childhood gluten could be directly caused/associated with your exposure maternal gluten intake. In fact, I would imagine these two factors in linked due to dietary similarities within families. Another possibility, instead of a time-varying covariate, is to simply start your maternal model follow-up time concordant to the child's follow-up period from 18 months. Hence, your mutual adjustment here would then discount the period of time where temporality was violated for baseline covariate adjustment. At the least, this analysis should be presented as a sensitivity analyses to stress test the results. 

4) Related to point 3) it would then be useful to quantify the relationship between maternal gluten intake and childhood gluten intake - if there is any association between these factors - as family dietary patterns should be inevitably correlated which would again suggest the two exposures are linked. 

5) Lines 163 - The authors adjusted for potential correlation between siblings in this cohort - which is good practice but is possible to identify within this cohort siblings? If this is possible, this presents a very unique opportunity to conduct a stratified analyses, comparing low and high gluten intake between siblings to account for the influence of genetics. Potential consideration for the discussion or future analyses. 

6) Any potential sub-group differences between child's gender and birthweight would be useful to see (perhaps in a KM survival plot)

Reviewer #2: The manuscript presents findings on type 1 diabetes outcome in children in relation to gluten intake by mothers during pregnancy and by children at age 18 months. The findings are from the Norwegian Mother and Child Cohort study (MoBa). Previous findings from a Danish study indicated that increased gluten intake during pregnancy was associated with increased risk in the offspring. The Moba study was unable to confirm the Danish findings. The authors show that gluten intake in mid-pregnancy was not associated with the development of type 1 diabetes in the child (adjusted HR, 1.02; 95%CI, 0.73-1.43). They also find that the child's gluten intake at age 18 months was associated with the development of type 1 diabetes (HR, 1.46 per 10 g/day increase). 

The study is important. It is generally sound and well performed. The question whether gluten intake influences type 1 diabetes risk remains unclear.

Comments to address:

1. An aspect of the analysis that requires some justification is the use of 10 g/day increase for the HRs. This is particularly so for the children where the average intake was 8.8 g/day (SD, 3.6) and the median 8.2 g/day. A 10 g/day increase is nearly 3 SD. What is the rationale for relating the data to such a large increase other than consistency with the Danish study (where it is also a concern)? One can see from table 2 that the increased risk is only seen at the upper centiles of gluten intake at 18 months. The authors should report the HRs in relation to a smaller increment, at least in the children intake analysis.

2. A distribution of the gluten intake in pregnancy and at age 18 months should be shown.

3. The data in Sup Table 6 shows information from Table 2 plus model 3 that includes adjustment for celiac disease in the child. Model 3 is an important one, especially given the recent publication by the TEDDY study that reports increased transglutaminase autoantibody risk in relation to gluten intake and the known association between type 1 diabetes and celiac disease. The authors should, therefore, add model 3 to table 2 and remove sup table 6. 

4. The authors should discuss the findings of the above-mentioned TEDDY study (transglutaminase autoantibody risk and gluten intake; JAMA 2019).

5. There is a slight attenuation of risk associated with child gluten intake in model 3. A sensitivity analysis with and without the cases of celiac disease should be performed.

6. Children who have a mother with type 1 diabetes or celiac disease were excluded. These are likely to include a relevant number of cases of type 1 diabetes. The authors should add as supplementary information an analysis in which these children are included.

7. It would be of interest to show the findings in relation to celiac disease. One appreciates that this may be the subject of a separate manuscript, but it is likely to be relevant to the current analysis and the authors should seriously consider this. 

8. Many of the cases of type 1 diabetes in the current study will have developed islet autoantibodies prior to age 18 months. The authors should discuss this aspect since their findings do not provide knowledge as to whether the association is related to risk of islet autoimmunity or development/progression to diabetes.

9. A statement on the power of the study in the pregnancy dataset and in the children dataset should be given. 

10. Some statistics (eg p values) should be given in the text in the second paragraph of the results when stating lower than. 

11. Although the food questionnaire is validated, it has its limitations and it is unlikely to be as robust as a 3-day record. Some comment on its limitation should be included.

Reviewer: Ezio Bonifacio

Reviewer #3: This interesting manuscript evaluates the relationship between gluten intake during pregnancy or early childhood and risk of developing TIDM during the first 10-20 years of life. The authors find a relationship between reported gluten intake at age 18 months and risk of subsequent TIDM development. Strengths of the study include the nationally representative sample, good sample size, robust identification of exposures and outcomes and limited missing data. Limitations include the lack of genetic information (given that genetic risk factors for TIDM are recognised), the lack of information on other dietary intake and whether or not this is also associated with TIDM risk, the lack of information on other autoimmune outcomes or TIDM-associated antibody levels to establish whether these are associated with gluten intake, and the lack of information on gluten intake at other timepoints during early childhood. It is hard to understand whether this association is specific to gluten when oat intake was not explored, or other foods which don't contain gluten.

[LINK]

---

## [Decision Letter · Decision Letter 1]

19 Dec 2019

Dear Dr. Lund-Blix,

Thank you very much for re-submitting your manuscript "Maternal and child gluten intake and risk of type 1 diabetes: The Norwegian Mother and Child Cohort Study" (PMEDICINE-D-19-03046R1) for review by PLOS Medicine.

I have discussed the paper with my colleagues and the academic editor and it was also seen again by xxx reviewers. I am pleased to say that provided the remaining editorial and production issues are dealt with we are planning to accept the paper for publication in the journal.

[LINK]

We look forward to receiving the revised manuscript by 23 December 2019. 

Sincerely,

Adya Misra, PhD

Senior Editor 

PLOS Medicine

plosmedicine.org

Requests from Editors:

Title – please remove ‘risk’ in favour of 'association' as this is an observational study

Data – the link provided appears to be a broken page https://www.fhi.no/en/op/dataaccess-from-health-registries-health-studies-and-biobanks/data-from-moba/mobaresearch-data-files/

Author summary – Please remove CIs as this is supposed to be a lay summary?

Please add a space between text and reference brackets throughout

Abstract-please clarify what is meant by mid-pregnancy?

Author summary- please change “may increase risk…” to “associated with increased risk..” as this is an observational study

Discussion- please avoid assertions of primacy 

Please present and organize the Discussion as follows: a short, clear summary of the article's findings; what the study adds to existing research and where and why the results may differ from previous research; strengths and limitations of the study; implications and next steps for research, clinical practice, and/or public policy; one-paragraph conclusion.

Please clarify why the KM plots requested by reviewers have not been included in the main manuscript or the supplementary information. Please include this as SI files and provide a reference within the manuscript

Did your study have a prospective protocol or analysis plan? Please state this (either way) early in the Methods section.

Comments from Reviewers:

Reviewer #1: The authors did an excellent job revising and amending their manuscript. They have thoroughly addressed my queries and provided additional analyses. With the additional analyses, their primary findings stand, hence this paper can be recommended for publication. As with all observational studies, there are limitations but the authors have done their best to control for these factors. 

The only final point two points I would suggest are in the discussion:

1) Limitations - no description of ethnicity but assume this is predominantly White Caucasian as gluten, both exposures with dietary influences and rates of T2DM can vary by ethnic group

2) Conclusions - authors suggest (Lines 388 onwards) that future interventional studies may be warranted to explore this in children but I think this may be difficult to actually test in practice - given that the long duration of follow-up, the ethics around restricting gluten in otherwise healthy children. I would think that more causal observational designs would be more likely and practical (i.e. Mendelian randomisation) with identifying genetics risk markers that are instrumental variables for gluten metabolism and then testing the relationship between the instrument which is not confounded by environmental influences on the outcome of T2DM. 

Reviewer #2: Thank you. No further comments.

[LINK]

---

## [Editor Report · Decision Letter 2]

27 Jan 2020

Dear Dr. Lund-Blix, 

On behalf of my colleagues and the academic editor, Dr. Ronald Ma, I am delighted to inform you that your manuscript entitled "Maternal and child gluten intake and association with type 1 diabetes: The Norwegian Mother and Child Cohort Study" (PMEDICINE-D-19-03046R2) has been accepted for publication in PLOS Medicine. 

PRODUCTION PROCESS

PRESS

PROFILE INFORMATION

Thank you again for submitting the manuscript to PLOS Medicine. We look forward to publishing it. 

Best wishes, 

Adya Misra, PhD

Senior Editor 

PLOS Medicine

plosmedicine.org